# Evaluation of Body Changes and the Anti-Obesity Effect after Consumption of Korean Fermented Food, Cheonggukjang: Randomized, Double-Blind Clinical Trial

**DOI:** 10.3390/foods12112190

**Published:** 2023-05-30

**Authors:** A Lum Han, Su-Ji Jeong, Myeong-Seon Ryu, Hee-Jong Yang, Do-Youn Jeong, Yoo-Bin Seo

**Affiliations:** 1Department of Family Medicine, Wonkwang University Hospital, Iksan 54538, Republic of Korea; 2Microbial Institute for Fermentation Industry, Sunchang 56048, Republic of Korea; yo217@naver.com (S.-J.J.); rms6223@naver.com (M.-S.R.); godfiltss@naver.com (H.-J.Y.); jdy2534@korea.kr (D.-Y.J.); 3Department of Family Medicine, Wonkwang University Sanbon Hospital, Sanbon 15865, Republic of Korea; ybseo610@naver.com

**Keywords:** *Bacillus* spp., cheonggukjang, obesity, short-chain fatty acids, biogenic amines

## Abstract

Cheonggukjang is a traditional Korean fermented soybean food with potential health benefits. For this reason, Cheonggukjang is consumed in the form of pills in addition to being used as a food ingredient. There are few clinical studies that have evaluated changes in various health indicators through blood and stool tests before and after consumption of Cheonggukjang. In this study, symptoms and hematological changes were analyzed before and after the intake of traditional Cheonggukjang pills containing high-dose (*n =* 19) or low-dose (*n =* 20) beneficial bacteria and commercial Cheonggukjang pills (*n =* 20). Anti-obesity effects and body composition changes were determined before and after Cheonggukjang consumption. Lastly, the changes in microorganisms and short-chain fatty acids in the stool were compared. No changes in obesity and inflammation-related indicators were observed before and after Cheonggukjang consumption. The *Firmicutes*/*Bacteroidetes* ratio, associated with obesity, decreased in all three groups after Cheonggukjang consumption, but no statistical significance was indicated. Cheonggukjang contained various BAs, but they did not adversely affect symptoms and hematological changes in the participants. BAs generated during the manufacturing process of Cheonggukjang did not have any adverse effects in this randomized, double-blind clinical trial. Further research is needed in future concerning the anti-obesity effect or regarding changes in the microbiome and short-chain fatty acids in feces.

## 1. Introduction

Soybeans are a food with high nutritional value, containing many vitamins and minerals [1]; they are used as a dietary source of nutrition in daily life in Asian countries, including Korea [2]. Cheonggukjang is a traditional Korean fermented food made from soybeans and is a nutritionally vital functional food because it contains essential amino acids, fatty acids, organic acids, minerals, and vitamins [2]. Cheonggukjang belongs to a family of traditional sauces, including soy sauce, soybean paste, and red pepper paste. 

The aging of Cheonggukjang involves a fermentation process using microorganisms, and strains of the genus *Bacillus*, such as *Bacillus subtilis*, *Bacillus licheniformis*, and *Bacillus megaterium*, used alone [3,4,5], or two strains are mixed [6]. During fermentation, *Bacillus* spp. break down the proteins and other nutrients in soybeans, producing a variety of compounds, including a sticky mucus-like substance. The mucus-like substance produced during Cheonggukjang fermentation is a complex mixture of various compounds, including levan and polyglutamate. However, harmful substances, such as biogenic amines (BAs), can be produced during fermentation. BAs are low-molecular-weight compounds produced by the amination and transamination reactions of aldehydes and ketones [7]. BAs are in the form of aliphatic (putrescine (PUT), cadaverine (CAD), spermine, spermidine (SPD)), aromatic (tyramine (TYR), phenylethylamine), and heterocyclic structures (histamine (HIS), tryptamine) [8]. BAs are an essential substance for maintaining the physiological function of the human body and are balanced in trace amounts through metabolic processes; however, BAs generated during the manufacturing process of food sometimes affect the human body. In particular, in soybean- or milk-containing foods with a high amino acid and protein content, BAs that exceed the digestion limit of the human body may be generated during the fermentation process. These BAs are continuously detected in various fermented foods, including Cheonggukjang, and can cause nervous and vascular system abnormalities in the human body. HIS can cause food poisoning, and some BAs can be converted into carcinogens such as N-nitrosamine [8]. Excessive HIS intake can lower blood pressure and cause allergies, TYR can increase blood pressure and cause headaches, and CAD and PUT can produce potent carcinogens, including nitrosamines, nitrosopiperidine, and nitrosopyrrolidine in the intestines [9,10,11]. Because of this risk, the European Union carried out a project to reduce BAs in traditional European fermented foods, such as cheese, wine, and sausages, from 2008 to 2011 [12]. Worldwide, studies have been conducted on the content of BAs in various foods, such as meat, fish, dairy products, wine, chocolate, vegetables, fruits, and nuts. As a standard for managing amines, tolerance standards for HIS and TYR have been set concerning fermented foods, marine products, and wine [12].

In Cheonggukjang, the tissue is softened by the action of microorganisms such as *B. subtilis* on steamed soybean. In addition, the mucilage produced during the fermentation process of Cheonggukjang is a biopolymer that is a mixture of gamma-polyglutamic acid and levan, a fructose polymer [13]. These substances in Cheonggukjang inhibit adipocyte differentiation and adipogenesis, proving their usefulness in obesity and metabolic disease treatment [14]. In previous clinical studies on Cheonggukjang, Cheonggukjang reduced body weight and body fat, decreased low-density lipoprotein cholesterol and total cholesterol levels, and improved serum lipids [15].

In this clinical trial, we first analyzed the bioamine content of Cheonggukjang, with a high or low content of effective microbial strains, and commercially available Cheonggukjang, and attempted to confirm its safety when consumed. The anti-obesity and lipid-lowering effects of the three types of Cheonggukjang were investigated. Lastly, microbial changes in stool samples after Cheonggukjang ingestion were compared and confirmed.

## 2. Materials and Methods

### 2.1. Study Design

This study was an 8-week randomized, double-blind clinical trial (registration number: KCT0007734). The participants were asked to visit the research center three times. After 4 weeks of Cheonggukjang supplementation, the participants visited the research center to check vital signs, side effects, and medication compliance. Efficacy indicators, such as weight, abdominal fat computed tomography (CT), inflammatory markers, lipid profile, and blood tests, were assessed at the first and last visits. The following Cheonggukjang tablets were provided to the participants: tablets made from traditional Cheonggukjang with a high useful microorganisms content (HTC), tablets made from traditional Cheonggukjang with a low useful microorganisms content (LTC), and pills made from commercially available Cheonggukjang (CC). Enrolled participants were randomly assigned to one of the three groups: HTC (*n =* 19), LTC (*n =* 20), or CC pills (*n =* 20). A total of 62 study participants were enrolled, and the dropout rate was 5%. Therefore, 59 participants completed the study.

Screening numbers were provided to those who provided written consent to participate in the study. The numbers represent the order of study participants and range from 01 to 62. The number assigned to each subject was used as the participant identification code. All participants were instructed not to take drugs or healthy functional foods other than the Cheonggukjang pills and were asked to maintain their usual diet and activity level.

### 2.2. BAs Analysis

BA quantification followed the Korean Food Code (MFDS, 2016). The final volume of the sample (1.0 mL) was adjusted to 50 mL with 0.1 N HCl to prepare the test solution. Then, 1 mL of test solution was placed in a test tube, and 0.1 mL of the internal standard solution, saturated sodium carbonate solution, and 0.8 mL of a 1% dansyl chloride solution were added and mixed thoroughly. The mixture was sealed and heated at 45 °C for 1 h to form a derivative. Then, 0.5 mL of 10% proline and 5 mL of ether solution were added to the derivatized standard solution, and the supernatant was collected and concentrated using nitrogen gas. The solution was then dissolved in 1 mL of acetonitrile and filtered through a 0.45 µm syringe filter. The filtered solution was analyzed using high-performance liquid chromatography (HPLC, Ultimate 3000, Thermo Fisher Scientific Co., Waltham, MA, USA). The analysis conditions are listed in Table 1.

### 2.3. Participants

Obese volunteers with a body mass index (BMI) ≥25 kg/m^2^ (age 19–70 years) were recruited and randomly assigned to three groups. The exclusion criteria were as follows: >10% change in body weight in the last 3 months, cardiovascular diseases such as arrhythmia, heart failure, myocardial infarction, and pacemaker use, allergy or hypersensitivity to any component of the test product, colon diseases such as Crohn’s disease, history of gastrointestinal surgery (e.g., appendix or bowel surgery), participation in another clinical trial within the last 2 months, liver dysfunction or acute/chronic kidney disease, antipsychotic drug therapy use within the past 2 months, laboratory test abnormality assessed by the investigator, abnormal psychological condition, alcohol or drug abuse history, and pregnancy or breastfeeding. The study protocol was approved by the Institutional Review Board of Wonkwang University Hospital (IRB approval number: WKUH 2022-05-025).

Participants were given a dietary diary and recorded all the foods they consumed as accurately as possible. The study participants completed a physical activity questionnaire based on the Global Physical Activity Questionnaire during their visit [16]. Participants ingested 3 g Cheonggukjang tablets (3 g fermented soybeans) once a day.

### 2.4. Safety Assessment

Participants underwent screening tests, including electrocardiogram, urinalysis, hematology, and blood chemistry tests. These included white and red blood cell counts and hemoglobin, hematocrit, platelet counts, total protein, albumin, alanine aminotransferase (ALT), aspartate aminotransferase (AST), blood urea nitrogen (BUN), and creatinine levels. The participants’ pulse and blood pressure were measured after a 10-min break at each visit. They reported any symptoms that occurred while taking the Cheonggukjang pills.

### 2.5. Indicators of Obesity Assessment

CT (HiSpeed CT/e; General Electric, Boston, MA, USA) was performed with the participant in a supine position for abdominal fat measurement. The total abdominal, subcutaneous, and visceral fat areas were measured and expressed in mm^2^. The participants fasted for >12 h before blood collection. Total cholesterol (TC), low-density lipoprotein cholesterol (LDL-C), high-density lipoprotein cholesterol (HDL-C), triglyceride (TG), nonHDL cholesterol (NonHDL-C), ALT, AST, gamma-glutamyl transferase (GGT), BUN, creatinine, glucose, insulin and high-sensitivity C-reactive protein levels were measured using a Hitachi 7600 automatic analyzer (Hitachi, Tokyo, Japan). HOMA_IR (Homeostatic Model Assessment for Insulin Resistance) and QUICKI (quantitative insulin-sensitivity check index) levels were calculated.

### 2.6. Inflammation Marker Assessment

Serum inflammatory markers, interleukin (IL)-6 and haptoglobin, were measured to determine whether inflammation improved before and after taking Cheonggukjang pills. Blood was left for 30 min for clotting after collection and then centrifuged at 3000 rpm for 10 min. The separated supernatant was transferred to a microtube, stored at −70 °C, collected, and analyzed by a commissioned institution (SCL Healthcare Co., Ltd., 13, Heungdeok 1-ro, Giheung-gu, Yongin-si, Gyeonggi-do, Republic of Korea). 

### 2.7. Changes in the Gut Microbiome Assessment

To analyze changes in the intestinal microbiome, the participants underwent a stool test at the first visit and 8 weeks later. Participants collected feces of 1 g or more using the MICROBE & ME stool collection kit (Macrogen, Seoul, Republic of Korea) and submitted it frozen. 

### 2.8. Experimental Cheonggukjang Pill Preparation

In traditional Cheonggukjang, rice straw is used as a fermentation starter. In the conventional method, soybeans are soaked and boiled until they become soft. The beans were then covered with rice straw and placed in a warm, humid environment for fermentation. The temperature is typically 35–40 °C, and the humidity is 80% to promote the microorganism growth for 24 to 72 h.

The manufacturing process of the traditional Cheonggukjang is shown in Figure 1. To create a tablet form of Cheonggukjang for clinical trials, Cheonggukjang was freeze-dried and crushed. Next, it was mixed with the excipients following the ratios specified in Table 2 before manufacturing.

### 2.9. Statistical Analysis

All statistical analyses were performed using PASW statistics 23 (previously SPSS statistics) (SPSS version 23.0; IMP SPSS, Chicago, IL, USA). All data are expressed as the mean ± standard error or percentages (%) for categorical variables. Values of *p* < 0.05 were considered significant.

The sample size was determined to achieve 80% statistical power with an alpha of 0.05. The sample size for each group was determined by allowing a dropout rate of 20%. Efficacy parameters were analyzed in the per protocol group, and safety parameters were analyzed in the intention-to-treat group. A chi-square test was performed to determine the baseline differences in the frequencies of categorized variables between the groups. Students’ paired t-test was performed to assess differences between the groups before and after the 8-week intervention period. A linear mixed-effects model was applied to the repeated measures data for each continuous outcome variable and data. An expert analyzed the 24-h dietary intake data using Can-Pro 3.0 software (Korean Nutrition Society, Seoul, Republic of Korea).

## 3. Results

### 3.1. Participants

A total of 62 participants were enrolled; one did not meet the inclusion criteria, and two declined to participate. The remaining 59 participants patients were randomly assigned to three groups: HTC, LTC, and CC, and all participants completed the 8-week study. A total of 59 participants from the HTC (*n =* 19), LTC (*n =* 20), and CC (*n =* 20) groups were included in the final analysis (Table 3). No participants indicated any adverse events after completing the study.

### 3.2. Anthropometric Parameters

The general characteristics of the participants are presented in Table 3. A cross-analysis was performed for sex, drinking, and smoking, and a one-way analysis of variance was performed for the age, weight, and BMI. There were no significant differences in the baseline characteristics among the three groups regarding age, sex, weight, height, drinking, smoking, initial weight, and initial BMI (*p* > 0.05). The dietary intake surveys found no significant changes in caloric intake within or between the groups (*p* > 0.05). The physical activity survey found no significant differences in the metabolic equivalents of tasks within or between the groups (*p* > 0.05).

The average deviation values for PUT, CAD, HIS, serotonin, TYR, SPD, noradrenaline, dopamine, and spermine were measured for the three types of soybean paste products. CAD, HIS, and SPD were detected in all samples among the nine biogenic amines tested, whereas, the other BAs were not. Although the content varied between the samples, these three BAs were consistently present (Table 4).

### 3.3. Safety Assessment

In the blood tests, such as liver inflammation level, kidney function, and general blood, there was no increase after Cheonggukjang pill administration. However, after consuming Cheonggukjang, the GGT, AST, and ALT levels decreased in the CC group. In all groups, no abnormal symptoms were reported after taking Cheonggukjang pills.

### 3.4. Effects on Obesity and Inflammation

Cheonggukjang showed no anti-obesity effects. In contrast, the body fat mass and fat percentage increased in the LTC group after administering Cheonggukjang pills. No change in the abdominal fat CT results in any group after Cheonggukjang administration was observed. Cheonggukjang had no effect on TC, LDL-C, HDL-C, TG, or NonHDL-C. Cheonggukjang did not affect glucose, insulin, HOMA_IR, or QUICKI levels. The serum inflammatory markers, hs-CRP, IL-6, and haptoglobin, also showed no change after Cheonggukjang administration in all groups. However, the GGT, AST, and ALT levels decreased in the CC group (Table 5). In all groups, no abnormal symptoms were reported after taking Cheonggukjang pills. 

### 3.5. Microbiome and Short-Chain Fatty Acids Analysis in Feces

The Firmicutes/Bacteroidetes (F/B) ratio, associated with obesity in all three groups, decreased after Cheonggukjang pill consumption, but there was no statistical significance. In all three groups, the presence of beneficial, harmful, and other microorganisms was evaluated in stool samples after consuming the Cheonggukjang pill. After supplementation, the numbers of beneficial and harmful bacteria did not show statistically significant differences in any of the three groups. The beneficial microorganisms included *Lactobacillus* spp., *Bifidobacterium* spp., *Lactococcus lactis*, *Enterococcus faecium*, and *Bacteroides* spp. Harmful bacteria included *Clostridium perfringens, Bacteroides eggerthii, Sutterella stercoricanis, Ruminococcus torques, Parabacteroides merdae,* and *Parabacteroides distasonis* (Table 6).

After supplementation, the short-chain fatty acid concentration in the feces decreased only in the HCT group, but the difference was not statistically significant. The short-chain fatty acids analyzed in the feces were acetic, propionic, and butyric acid (Table 7).

## 4. Discussion

Cheonggukjang is a traditional Korean food made from fermented boiled soybeans and rice straw. Cheonggukjang is a functional food because it contains various biologically active substances, including isoflavones, phytic acid, saponins, trypsin inhibitors, tocopherols, unsaturated fatty acids, dietary fiber, oligosaccharides, antioxidants, and thrombolytic enzymes [17]. These components are essential because of their potential health benefits. 

BAs, produced during soy protein fermentation (the main ingredient in fermented soybean products such as Cheonggukjang), affect cell growth and proliferation. However, BAs can also cause adverse effects, such as skin inflammation, headaches, and abdominal pain, and may be carcinogenic through intestinal microbial metabolism [18]. Additionally, BAs can stimulate the blood vessels and nerves, causing discomfort in some individuals. Therefore, the BA level in food has been used to indicate food safety, which is also considered a quality indicator for evaluating good manufacturing practices [19]. Producing many fermented foods worldwide is a complex biochemical process involving several microbial species that play a crucial role in BA accumulation [19,20]. Therefore, we investigated the BA content in Cheonggukjang prepared traditionally and in fermented foods. In addition, human safety was confirmed by examining the BA content in conventional Cheonggukjang with many and fewer effective strains.

Food microorganisms typically produce BA by secreting amino acid decarboxylases. Therefore, attempts have been made to characterize the microbiomes of fermented foods and evaluate their ability to produce BAs to gain better insight into how BA accumulates in food [18,19,20]. Controlling the BA content will ensure the quality and safety of such popular traditional fermented foods as Cheonggukjang. Therefore, it is necessary to investigate the BA content using the existing fermented food manufacturing process and confirm that it is clinically safe for human consumption. However, despite the regular consumption of traditional fermented products, i.e., Cheonggukjang, no cases of illness or death caused by BA in these fermented foods have been reported. Simultaneously, research continues to produce evidence of the potential health benefits of consuming these traditional fermented foods [21,22,23,24,25,26]. 

A previous study conducted in Korea investigated the contents of eight BAs and 18 inorganic components in 37 Cheonggukjang products distributed in supermarkets and markets. Among the eight BA components, the TYR content was the highest at 113.7 mg/kg, and the concentration difference was large depending on the sample [27]. Many types of Cheonggukjang are produced in Korea, and each product has a marked difference in flavor and nutritional content. This difference is because each region uses different raw soybean varieties and fermentation processes, which are not uniformized and systematized [27]. Therefore, comparing the BA content in various types of Cheonggukjang is meaningless because the type and BA content vary depending on the factors influencing Cheonggukjang production. 

However, research on the effect of BAs on the human body is needed. In Korea, studies have been conducted to analyze eight BAs, namely TRP, PHE, PUT, CAD, HIS, TYR, SPD, and SPM, in various Cheonggukjang products. Different results were obtained for each study, and in the same study, multiple values were found depending on the Cheonggukjang product [28,29,30]. Among the analyzed BAs, substances related to risk were PUT, CAD, HIS, TYR, and PHE, but no risk-related results have been reported for other amines [30]. In animal models, Cheonggukjang had various metabolic benefits, including anti-obesity, anti-diabetic, anti-inflammatory, and neuroprotective effects [31,32,33]. Cheonggukjang reduced body weight, epididymal fat accumulation, serum total cholesterol, and low-density lipoprotein cholesterol in an animal model of diet-induced obesity [31]. In addition, Cheonggukjang supplementation substantially reduced blood glucose and glycated hemoglobin levels in a mouse model and improved insulin resistance [32]. However, our study showed no improvement in all obesity-related indicators, such as lipid profile, blood glucose, insulin index, abdominal visceral fat CT, and body composition analysis results after Cheonggukjang pill administration. Previous animal models showed the anti-inflammatory effect of Cheonggukjang [31,32], but in our study, the anti-inflammatory indices hs-CRP, haptoglobin, and IL-6 did not improve after Cheonggukjang pill administration.

The *Firmicutes* phylum accounts for most of the intestinal microbiota. A decrease in *Firmicutes* and an increase in *Bacteroidetes* were associated with weight loss [34]. *Bacillus, Clostridium, Lactococcus,* and *Streptococcus* make up the majority of the *Firmicutes* phylum and the rest of the phylum is mostly *Bacteroides acidifaciens, B. distasonis,* and *B. fragilis* [35]. This study only investigated the two major bacterial phyla of the gastrointestinal tract, *Firmicutes*, and *Bacteroidetes*, to simplify the complex problems of the total microbial components. The F/B ratio is associated with homeostasis, and a change can cause various pathological conditions. For example, an increase in this ratio leads to obesity [36]. In this study, the F/B ratio of all three groups who ingested the Cheonggukjang pills decreased, but the difference was not statistically significant. An analysis of stool samples from Japanese people with obesity showed a substantial difference in the F/B ratio. The percentage of *Firmicutes* was 37.0 ± 9.1% (lean participants) and 40.8 ± 15.0% (obese participants), but the percentage of *Bacteroidetes* was 44.0 ± 9.8% (lean participants) and 37.0 ± 14.0% (obese participants) [37]. Researchers reported similar results for the F/B ratio related to BMI in 61 Ukrainian adults. Individuals with an F/B ratio of ≥1 were more likely to be overweight than those with < 1 [38]. Similarly, Qatar patients (37 obese and 36 not obese people) had an F/B ratio of 2.25 ± 1.83 and 1.76 ± 0.58, respectively [39]. In Kazakhstan [40] and Belgium [41], the F/B ratio in the obesity group was relatively high compared to that in the control group of schoolchildren. In contrast, no relationship between the F/B ratio and weight gain or BMI was observed. The fecal samples collected from obese patients (BMI ≥ 30 kg/m^2^) and not obese patients (BMI < 25 kg/m^2^) had no statistical difference in the F/B ratio in Korean youth (13–16 years) (0.50 ± 0.53 in not obese vs. 0.56 ± 0.86 in obese people). The *Firmicutes* to *Bacteroidetes* ratio is a common index to measure the structure of the gut microbiota. In research at the phylum level, an increase in the *Firmicutes* to *Bacteroidetes* ratio was observed in overweight/obese adults, resulting in an increased abundance of *Firmicutes* and a reduced abundance of *Bacteroidetes*. However, members of *Firmicutes* show greater heterogeneity in their composition than *Bacteroidetes* in the gut of metabolic syndrome patients (e.g., overweight/obesity), especially in the gut of children. For example, *Clostridium* (*Firmicutes* phylum) is positively associated with the body mass index (BMI) in children, and it is more important in young adults. An increased abundance of *Clostridium*, *Dorea*, and *Ruminococcus* (*Firmicutes* phylum) was also detected in Irritable Bowel Syndrome.

However, this study had several limitations. First, although the participants were instructed not to change their eating and exercise habits, the diet and activity levels could not be fully controlled. Second, the sample size of the participants was small and could not be generalized.

## 5. Conclusions

There has never been a clinical study that simultaneously confirmed various indicators such as safety, anti-obesity-related indicators, inflammation indicators, microbiome, and short-chain fatty acids after consuming Cheonggukjang. Although the indicators did not have statistical significance, these research results are worth referencing for future clinical trials on Cheonggukjang consumption. The results show no particular change in the indicators; therefore, there was no anti-obesity effect of Cheonggukjang. In addition, although there was no statistical significance, the F/B ratio decreased in all participants in the three groups, giving value to further research.

## Figures and Tables

**Figure 1 foods-12-02190-f001:**
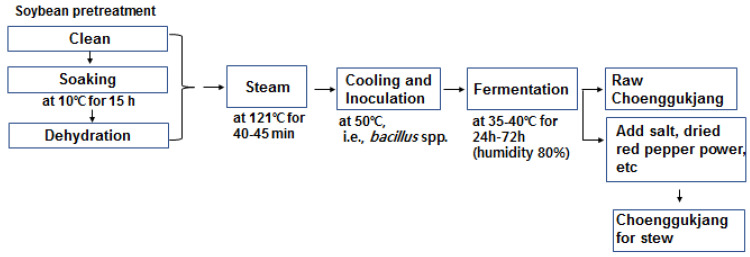
The manufacturing process of traditional Cheonggukjang.

**Table 1 foods-12-02190-t001:** Operation conditions of HPLC for biogenic amines analysis.

**Instrument**	Agilent 1200 series (Agilent Technologies, Santa Clara, CA, USA)
**Column**	CapcellPak C18 column
**Detector**	DAD detector (254 nm)
**Mobile phase**	A: 0.1% formic acid in H_2_O B: 0.1% formic acid in ACN
**Gradients condition**	A:B = 45:55, 0~10 min A:B = 35:65, 10~15 min A:B = 20:80, 15~20 min A:B = 10:90, 20~30 min A:B = 10:90, >40 min
**Flow rate**	1 mL/min
**Temperature**	40 °C
**Injection volume**	20 µL

**Table 2 foods-12-02190-t002:** Composition of the Cheonggukjang pills’ ingredients.

	HTC		LTC		CC	
	Content (g)	Ratio (%)	Content (g)	Ratio (%)	Content (g)	Ratio (%)
Freeze-dried Cheonggukjang powder	2.97	90	2.97	90	2.97	90
Glutinous rice flour	0.33	10	0.33	10	0.33	10
Total	3	100	3	100	3	100

HTC, traditional Cheonggukjang containing a high dose of beneficial microbes; LTC, traditional Cheonggukjang containing a low dose of effective microbes; CC, commercially prepared Cheonggukjang.

**Table 3 foods-12-02190-t003:** General characteristics of participants.

Value	Group			
	HTC (*n =* 19)	LTC (*n =* 20)	CC (*n =* 20)	*p*-Value
Sex (M/F)	9/10	10/10	11/7	0.675
Drinking (n)	10 (52.6)	7 (35.0)	11 (61.1)	0.256
Smoking (n)	2 (10.5)	2 (10.0)	3 (16.7)	0.790
Age	38.00 ± 10.52	41.80 ± 12.64	41.80 ± 12.64	0.213
Weight	78.33 ± 12.98	75.31 ± 12.97	80.74 ± 14.15	0.458
BMI	28.17 ± 3.39	27.98 ± 2.91	28.25 ± 2.51	0.958

HTC, traditional Cheonggukjang containing a high dose of beneficial microbes; LTC, traditional Cheonggukjang containing a low dose of effective microbes; CC, commercially prepared Cheonggukjang; M, male; F, female; BMI, body mass index. Values are presented as the mean ± standard deviation or number (percentage).

**Table 4 foods-12-02190-t004:** Biogenic amine contents in Cheonggukjang products.

Sample	Biogenic Amine (mg/kg)									Total Contents (mg/kg)
	PUT	CAD	HIS	SER	TYR	SPD	NOR	DOP	SPM	
HTC	ND ^(1)^	96.28 ± 2.7 ^(2)^	619.34 ± 1.5	ND	ND	56.35 ± 1.1	ND	ND	ND	771.91 ± 5.0
LTC	ND	207.75 ± 2.7	91.35 ± 0.9	ND	ND	40.26 ± 0.2	ND	ND	ND	339.36 ± 3.2
CC	ND	164.55 ± 1.2	157.87 ± 1.1	ND	ND	43.80 ± 1.9	ND	ND	ND	366.23 ± 2.8

HTC, traditional Cheonggukjang containing a high dose of beneficial microbes; LTC, traditional Cheonggukjang containing a low dose of effective microbes; CC, commercially prepared Cheonggukjang; PUT, putrescine; CAD, cadaverine; HIS, histamine; SER, serotonin; TYR, tyramine; SPD, spermidine; NOR, noradrenaline; DOP, dopamine; SPM, spermine. ^(1)^ Not Detected. ^(2)^ mean ± SD (standard deviation).

**Table 5 foods-12-02190-t005:** Efficacy evaluation between the three groups.

Value	Group								
	HTC (*n =* 19)			LTC (*n =* 20)			CC (*n =* 20)		
	Before	After	*p*-Value	Before	After	*p*- Value	Before	After	*p*- Value
HC	103.03 ± 6.64	102.85 ± 6.56	0.128	101.33 ± 6.25	101.04 ± 5.95	0.429	103.42 ± 5.57	103.31 ± 5.39	0.497
WHR	0.91 ± 0.05	0.91 ± 0.05	0.871	0.91 ± 0.05	1.01 ± 0.45	0.330	0.91 ± 0.05	0.92 ± 0.05	0.96
TC	203.21 ± 25.83	209.37 ± 27.1	0.239	206.25 ± 26.54	208.9 ± 29.05	0.686	207.17 ± 27.9	204.11 ± 31.04	0.363
LDL-C	121.79 ± 29.37	125.05 ± 32.91	0.563	132.45 ± 22.6	131.05 ± 22.93	0.809	129.28 ± 23.3	126.39 ± 27.32	0.494
HDL-C	51.89 ± 13.14	54.32 ± 13.9	0.208	47.1 ± 11.12	49 ± 9.82	0.235	48.78 ± 14.38	50.39 ± 11.41	0.384
Triglyceride	130.21 ± 111.54	127.95 ± 122.36	0.918	132.95 ± 69.38	134.1 ± 52.85	0.906	155.06 ± 97.32	134 ± 84.13	0.304
NonHDL-C	151.32 ± 30.54	155.05 ± 32.06	0.394	159.15 ± 26.22	159.9 ± 29.61	0.901	158.39 ± 28.85	153.72 ± 30.33	0.214
WC	93.79 ± 7.07	93.89 ± 6.99	0.361	92.3 ± 7.61	92.14 ± 7.49	0.762	94.08 ± 7.8	94.78 ± 7.5	0.125
IL-6	1.70 ± 0.72	1.82 ± 0.78	0.514	1.91 ± 1.02	1.88 ± 0.84	0.892	2.21 ± 1.67	1.82 ± 0.97	0.292
Haptoglobin	103.68 ± 48.32	100.21 ± 45.00	0.252	113.65 ± 51.17	112.15 ± 51.99	0.834	95.72 ± 59.49	102.00 ± 47.35	0.431
VF	128.8 ± 64.71	122.91 ± 68.34	0.176	135.8 ± 55.43	137.84 ± 53.56	0.690	143.85 ± 83.68	141.12 ± 80.54	0.601
SF	229.37 ± 86.49	222.47 ± 85.38	0.218	217.55 ± 69.03	213.35 ± 68.25	0.381	203.87 ± 65.9	211.1 ± 68.75	0.242
V/S	0.65 ± 0.49	0.64 ± 0.51	0.799	0.67 ± 0.35	0.68 ± 0.28	0.821	0.75 ± 0.54	0.7 ± 0.49	0.283
LAP	47.75 ± 33.16	48.3 ± 39.85	0.931	47.81 ± 25.81	48.37 ± 26.36	0.854	56.81 ± 32.11	49.28 ± 28.55	0.219
VAI	1.93 ± 1.56	2.08 ± 2.4	0.715	2.13 ± 1.38	2.05 ± 1.36	0.642	2.37 ± 1.6	1.96 ± 1.34	0.156
GGT	29.63 ± 18.59	29.32 ± 21.56	0.828	28.85 ± 16.45	27.6 ± 16.97	0.642	35.28 ± 27.18	30.17 ± 20.2	0.048
AST	25.11 ± 11.81	24.79 ± 11.53	0.860	24.5 ± 6.18	26.1 ± 6.66	0.208	27.28 ± 13.79	23.72 ± 8.81	0.049
ALT	24.89 ± 20.92	24.42 ± 23.03	0.827	26.8 ± 14.53	28.05 ± 15.27	0.585	31.33 ± 24.53	26.72 ± 22.03	0.030
BUN	12.74 ± 3.28	12.06 ± 2.99	0.370	12.85 ± 2.87	11.92 ± 2.39	0.172	12.92 ± 3.59	13.49 ± 3.27	0.311
Cr	0.83 ± 0.17	0.82 ± 0.18	0.277	0.83 ± 0.14	0.83 ± 0.14	0.970	0.84 ± 0.2	0.9 ± 0.3	0.324
Uric acid	6 ± 1.74	5.78 ± 1.43	0.162	5.88 ± 1.47	5.87 ± 1.52	0.976	6.59 ± 1.71	6.76 ± 1.68	0.390
Glucose	101.11 ± 7.32	101.21 ± 6.4	0.945	103.6 ± 8.44	105.7 ± 10.6	0.110	101.56 ± 9.26	103.06 ± 9.38	0.433
Insulin	8.91 ± 4.8	8.31 ± 4.54	0.376	8.58 ± 5.87	7.28 ± 2.91	0.190	11.67 ± 9.49	9.06 ± 5.39	0.051
HOMA_IR	2.25 ± 1.23	2.08 ± 1.15	0.356	2.19 ± 1.44	1.89 ± 0.74	0.252	3.03 ± 2.63	2.34 ± 1.43	0.075
QUICKI	0.35 ± 0.04	0.35 ± 0.04	0.606	0.35 ± 0.02	0.35 ± 0.02	0.443	0.35 ± 0.05	0.35 ± 0.04	0.629
hs-CRP	1.19 ± 0.83	2.56 ± 6.19	0.332	2.14 ± 2.27	1.78 ± 1.55	0.287	1.65 ± 1.9	2.7 ± 4.18	0.282

HTC, traditional Cheonggukjang containing a high dose of beneficial microbes; LTC, traditional Cheonggukjang containing a low dose of effective microbes; CC, commercially prepared Cheonggukjang; HC, hip circumference; WC, waist circumference; WHR, waist-hip ratio; TC, total cholesterol; LDL-C, low-density cholesterol; HDL-C, high-density cholesterol; TG, triglycerides; NonHDL-C, NonHDL-Cholesterol; VF, visceral fat; SF, subcutaneous fat; V/S, visceral fat/subcutaneous fat ratio; GGT, gamma-GT; AST, aspartate transaminase; ALT, alanine transaminase; BUN, blood urea nitrogen; Cr, creatinine; glucose; hs-CRP, high-sensitivity C-reactive protein. Values are presented as the mean ± SD.

**Table 6 foods-12-02190-t006:** Microbiome analysis of feces.

Value	HTC			LTC			CC		
	Before	After	*p*-Value	Before	After	*p*-Value	Before	After	*p*-Value
Firmicutes (%)	60.91 ± 9.79	64.96 ± 11.40	0.235	62.65 ± 11.74	60.40 ± 9.05	0.511	63.06 ± 10.80	62.11 ± 10.61	0.785
Bacteroidetes (%)	28.06 ± 12.79	24.34 ± 14.41	0.864	24.12 ± 12.45	29.69 ± 9.07	0.123	24.73 ± 14.77	26.87 ± 11.35	0.621
F/B	2.17	2.67		2.60	2.03		2.55	2.31	
Beneficial Bacteria	28.52 ± 10.50	31.30 ± 11.87	0.061	31.90 ± 11.02	29.96 ± 11.57	0.230	33.70 ± 7.69	31.80 ± 9.70	0.411
Harmful Bacteria	3.10 ± 2.47	3.43 ± 2.73	0.615	2.87 ± 2.91	2.44 ± 1.66	0.517	2.92 ± 2.41	3.44 ± 1.37	0.301
Others	68.38 ± 11.69	65.26 ± 13.20	0.063	65.23 ± 11.18	67.59 ± 11.66	0.172	63.38 ± 9.22	64.76 ± 10.13	0.570

HTC, traditional Cheonggukjang containing a high dose of beneficial microbes; LTC, traditional Cheonggukjang containing a low dose of effective microbes; CC, commercially prepared Cheonggukjang.

**Table 7 foods-12-02190-t007:** The concentration of short-chain fatty acids in feces.

Acid	HTC			LTC			CC		
	Before	After	*p*-Value	Before	After	*p*-Value	Before	After	*p*-Value
Acetic	46.66 ± 34.3	33.1 ± 26.59	0.123	49.39 ± 30.32	40.68 ± 24.83	0.341	51.98 ± 29.4	57.90 ± 47.81	0.568
Propionic	34.61 ± 25.49	26.97 ± 31.54	0.299	28.79 ± 18.45	20.85 ± 14.27	0.128	36.14 ± 24.49	35.00 ± 23.40	0.824
Butyric	26.36 ± 15.81	24.05 ± 18.57	0.639	26.58 ± 16.68	26.63 ± 15.67	0.993	27.86 ± 14.79	36.34 ± 26.29	0.232
Total	107.63 ± 71.70	84.12 ± 73.88	0.228	104.76 ± 59.79	88.16 ± 46.84	0.369	115.97 ± 62.06	129.23 ± 90.81	0.508

HTC, traditional Cheonggukjang containing a high dose of beneficial microbes; LTC, traditional Cheonggukjang containing a low dose of effective microbes; CC, commercially prepared Cheonggukjang.

## Data Availability

The data presented in this study are available on request from the corresponding author. The data are not publicly available, because the ownership of the data resides with the research institute.

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
