# Peer review of "Evaluation of Body Changes and the Anti-Obesity Effect after Consumption of Korean Fermented Food, Cheonggukjang: Randomized, Double-Blind Clinical Trial"

_foods, 2023, doi:10.3390/foods12112190_

Round 1

Reviewer 1 Report

The paper is interesting and well-written. In my opinion, the title should be modified because the work covers many aspects related to the consumption of Cheonggukjang.

Considering that the presented work concerns research conducted for 8 weeks, on a small group of people, the abstract is written too generally.

Author Response

Thank you very much for commenting on my research and improving the quality of my research.

Thanks to your point of view, the quality of my existing manuscript can be improved, so I am very grateful. The revised part is marked in red. Since I did my best to faithfully follow and respond to your comments and change the existing manuscript, I hope you will understand and consider this.

Reviewer 2 Report

Reviewer’s comments

The aim of this paper is to evaluate the biogenic amines (BAs) produced during fermentation and safety confirmation of Cheonggukjang contained various BAs when consumed in an 8-week randomized, double-blind clinical trial. This is a very interesting topic. I have thoroughly and carefully evaluated the manuscript. This manuscript looks scientifically with robust data. Nevertheless, there are some comments regarding the submitted manuscript:

My comments:

Title should be: “Evaluation of biogenic amines and the anti-obesity effect of Korean fermented food, Cheonggukjang: Randomized, double-blind clinical trial.”

Major comments:

Discussion: according to the latest publication (https://doi.org/10.3390/molecules28073213) I quote: The Firmicutes to Bacteroidetes ratio was a common index to measure the structure of the gut microbiota. In research at the phylum level, an increase in the Firmicutes to Bacteroidetes ratio was observed in overweight/obese adults, resulting in an increased abundance of Firmicutes and reduced abundance of Bacteroidetes. However, members of Firmicutes show greater heterogeneity in their composition than Bacteroidetes in the gut of metabolic syndrome patients (e.g., overweight/obesity), especially in the gut of children. For example, Clostridium (Firmicutes phylum) is positively associated with the body mass index (BMI) in children, and it is more important in young adults. An increased abundance of Clostridium, Dorea, and Ruminococcus (Firmicutes phylum) was also detected in Irritable Bowel Syndrome (IBS). (…) Thus, an analysis of microbiota at the phylum level/ratio is too general to notice differences in the microbiota composition in people with health problems.” It is necessary to include this issue in the discussion.

Minor comments

Lines 13-15: Please edit these sentences so that you do not repeat "This study" in each of them. Additionally, in this study, there was confirmation of the safety of Cheonggukjang contained various BAs. The study of isolated BAs was not conducted here. Please correct it.

Line 17:  Chungkukjang - a new name has been introduced without explaining that it is a synonym. Please put the synonyms (Chungkukjang) in brackets after the first use of the name Cheonggukjang. If there are other synonyms, please also put them in brackets.

Lines 26-26: Please change from “Cheonggukjang has anti-obesity potential..” to “Cheonggukjang is considered as anti-obesity potential” The authors have not proven that this product has anti-obesity properties

Lines 38-40: Please add a reference to where the authors got their knowledge about the richness of nutrients in this product

Line 81: After the first use of the full name of the microorganism (genus and species), the abbreviated name should be used in subsequent sentences, e.g. B. subtilis

Line 208: Please change the word from “excipient” to “excipients”

Lines 357-358: Please change the sentence from “Controlling the BA content will ensure the quality and safety of popular traditional fermented foods such as Cheonggukjang.” to “Controlling the BA content will ensure the quality and safety of such popular traditional fermented foods as Cheonggukjang.”

Line 362 Please change from “ such as” to i.e.

Line 404-406. Please write only microorganism names in italic.

Fig. 1. Please change the word from “dride” to “dried”

Author Response

(The authors gave the same response as above.)

Reviewer 3 Report

Comments to the Manuscript ID: foods-2332248

Title: Evaluation of biogenic amines and the anti-obesity effect of Korean fermented food, Cheonggukjang

 Title and aim

The title and aim of the article are incorrectly formulated.

In practice, as far as BAs research is concerned, it is sufficient to determine how many of them were formed in one or another product (‚BAs are toxic when consumed in excess‘ - line 60) and to assess whether the concentrations do not exceed the permissible norms.

Main aim can be: Changes in symptoms and hematology were analyzed before and after consuming Chungkukjang pills, commercial (n = 20), or traditional Cheonggukjang containing high-dose (n = 19) or low-dose (n = 20) beneficial microorganisms. Determination of anti-obesity effects and changes of fatty acids in stool before and after Cheonggukjang consumption etc.

The amount of BAs should be evaluated simply as a regulated component of the chemical composition, a parameter of food safety. Due to the concentrations of BAs, optimization of fermentation process with Bacillus spp. for Cheonggukjang production should be done to produce the possible lowest safe BAs concentrations.

This should also be reflected in the title. It can be : “Evaluation of the anti-obesity effect of Korean fermented food, Cheonggukjang, with low and high dose of Bacillus spp.”

Analysis of BAs – additional experiments

Abstract

Should be rewritten with greater emphasis on novelty and important research experiments.

I think that the aspect of novelty in the research is the mucus-like substance production during Cheonggukjang fermentation, including levan and polyglutamate, their concentrations, relation to the obesity reduction.

Keywords: Bacillus spp.; Cheonggukjang; obesity; short-chain fatty acids; biogenic amines.

Introduction.

Line 60. Please indicate what amounts of Bas in food are characterised as toxic.

Methods

Line 122. Please specify LOD and LOQ of the method

line 203. Specify, for what time concretely test products were fermented. Which microorganisms were used for different samples (high and low microorganisms content, commercial product? What are the differences?)

Line 216. Please indicate how many times chemical analyses have been performed.

Results. Please revise the description of the description, it lacks in-depth discussion to support the significance of the results. Sub-titles of sections need to be rethought and specified to reflect the main results.

 3.1 and 3.2 sections can be combined

Line 261. Add a new sub-title for the second section: Safety characteristics of Cheonggukjang products

Line 274. Remove sub-title

Lines 275-279. A reference to where the results of this section are presented is required.

Line 277. Change ‘administration’ to ‘consumption’

Line 277-278. Provide a broader explanation of what it means that the GGT, AST, and ALT levels decreased.

Line 280. Inflammation of what?

Line 292. Specify the title of Table 5. What is Efficacy evaluation??

Line 281. The title of your article contains the statement „Evaluation of biogenic amines and the anti-obesity effect of Korean fermented food", but you report that Cheonggukjang showed no anti-obesity effects? In the abstract you wrote that „The Firmicutes/Bacteroidetes ratio, associated with obesity, decreased in all three groups after Cheonggukjang consumption, but no statistical significance was indicated. Please write more clearly these statements in results section.

Line 402. For Firmicutes phylum use Italic font.

Conclusions. The conclusions you include in the paper should be more succinct and precise. Rather than repeating parts of the paper, use this last paragraph to wrap up what you want readers to remember most.

Author Response

(The authors gave the same response as above.)

Round 2

Reviewer 3 Report

I suggest accepting the manuscript for publication in Foods after an indication of the novelty of the results in the conclusions.

Author Response

Auther answer to reviewer 3 (Manuscript ID: foods-2332248)

Thank you very much for commenting on my research and improving the quality of my research.

Thanks to your point of view, the quality of my existing manuscript can be improved, so I am very grateful. The revised part is marked in red. Since I did my best to faithfully follow and respond to your comments and change the existing manuscript, I hope you will understand and consider this.

I suggest accepting the manuscript for publication in Foods after an indication of the novelty of the results in the conclusions.

  • I revised it as you mentioned.
